# Glycaemic Variability and Hyperglycaemia as Prognostic Markers of Major Cardiovascular Events in Diabetic Patients Hospitalised in Cardiology Intensive Care Unit for Acute Heart Failure

**DOI:** 10.3390/jcm11061549

**Published:** 2022-03-11

**Authors:** Edouard Gerbaud, Ambroise Bouchard de La Poterie, Thomas Baudinet, Michel Montaudon, Marie-Christine Beauvieux, Anne-Iris Lemaître, Laura Cetran, Benjamin Seguy, François Picard, Fritz-Line Vélayoudom, Alexandre Ouattara, Rémi Kabore, Pierre Coste, Pierre Domingues-Dos-Santos, Bogdan Catargi

**Affiliations:** 1Cardiology Intensive Care Unit and Interventional Cardiology, Hôpital Cardiologique du Haut-Lévêque, 33604 Pessac, France; ambroisebdlp@gmail.com (A.B.d.L.P.); thomas.baudinet@chu-bordeaux.fr (T.B.); laura.cetran@chu-bordeaux.fr (L.C.); benjamin.seguy@chu-bordeaux.fr (B.S.); pierre.coste@u-bordeaux.fr (P.C.); 2Bordeaux Cardio-Thoracic Research Centre, U1045, Bordeaux University, 33076 Bordeaux, France; michel.montaudon@chu-bordeaux.fr (M.M.); pierre.domingues-dos-santos@u-bordeaux.fr (P.D.-D.-S.); 3Biochemistry Laboratory, Hôpital Cardiologique du Haut-Lévêque, Bordeaux University, 33600 Pessac, France; marie-christine.beauvieux@chu-bordeaux.fr; 4Centre de Résonance Magnétique des Systèmes Biologiques, UMR 5536, CNRS, Bordeaux University, 33076 Bordeaux, France; 5Advanced Heart Failure Unit, Department of Cardiovascular Medicine, Hôpital Cardiologique du Haut-Lévêque, Bordeaux University, 33604 Pessac, France; anneiris.lemaitre@gmail.com (A.-I.L.); francois.picard@chu-bordeaux.fr (F.P.); 6Department of Diabetology-Endocrinology, University Hospital of Guadeloupe, 97159 Pointe-à-Pitre, France; fritz-line.velayoudom@univ-antilles.fr; 7Inserm UMR 1283, European Genomic Institute for Diabetes (EGID), Institut Pasteur de Lille, 59000 Lille, France; 8Department of Anesthesia and Critical Care, Magellan Medico-Surgical Centre, Bordeaux University, 33600 Pessac, France; alexandre.ouattara@chu-bordeaux.fr; 9Biology of Cardiovascular Diseases Centre, U1034, Bordeaux University, 33600 Pessac, France; 10Institut de Santé Publique, d’Épidémiologie et de Développement (ISPED), Bordeaux Population Health Research, U1219, Bordeaux University, 33000 Bordeaux, France; remi.kabore@u-bordeaux.fr; 11Institut de Rythmologie et Modélisation Cardiaque (IHU Liryc), Fondation Bordeaux Université, 33600 Pessac, France; 12Endocrinology-Metabolic Diseases, Hôpital Saint-André, Bordeaux University, 33000 Bordeaux, France; bogdan.catargi@chu-bordeaux.fr

**Keywords:** diabetes, glycaemic variability, acute heart failure, major adverse cardiovascular event

## Abstract

(1) Background: Hyperglycaemia and hypoglycaemia are both emerging risk factors for cardiovascular disease. Nevertheless, the potential effect of glycaemic variability (GV) on mid-term major cardiovascular events (MACE) in diabetic patients presenting with acute heart failure (AHF) remains unclear. This study investigates the prognostic value of GV in diabetic patients presenting with acute heart failure (AHF). (2) Methods: this was an observational study including consecutive patients with diabetes and AHF between January 2015 and November 2016. GV was calculated using standard deviation of glycaemia values during initial hospitalisation in the intensive cardiac care unit. MACE, including recurrent AHF, new-onset myocardial infarction, ischaemic stroke and cardiac death, were recorded. The predictive effects of GV on patient outcomes were analysed with respect to baseline characteristics and cardiac status. (3) Results: In total, 392 patients with diabetes and AHF were enrolled. During follow-up (median (interquartile range) 29 (6–51) months), MACE occurred in 227 patients (57.9%). In total, 92 patients died of cardiac causes (23.5%), 107 were hospitalised for heart failure (27.3%), 19 had new-onset myocardial infarction (4.8%) and 9 (2.3%) had an ischaemic stroke. Multivariable logistic regression analysis showed that GV > 50 mg/dL (2.70 mmol/L), age > 75 years, reduced left ventricular ejection fraction (LVEF < 30%) and female gender were independent predictors of MACE: hazard ratios (HR) of 3.16 (2.25–4.43; *p* < 0.001), 1.54 (1.14–2.08; *p* = 0.005), 1.47 (1.06–2.07; *p* = 0.02) and 1.43 (1.05–1.94; *p* = 0.03), respectively. (4) Conclusions: among other well-known factors of HF, a GV cut-off value of >50 mg/dL was the strongest independent predictive factor for mid-term MACE in patients with diabetes and AHF.

## 1. Introduction

Diabetes and chronic heart failure (CHF) are common comorbidities. Around 40% of patients managed for acute heart failure (AHF) had diabetes [1]. Diabetes can precipitate HF [2] and is associated with a worse prognosis of CHF [3]. Long-term control of diabetes, reflected by HbA_1c_, is an independent predictor of CHF [3]. A new concept, glycaemic variability (GV) [4,5], has emerged, and is associated with chronic diabetes complications such as microangiopathy [6] and macroangiopathy [7]. GV corresponds to fluctuations in blood glucose levels over a given interval of time [8]. The mechanisms underlying the deleterious effects of GV involve short-term fluctuations, inducing endothelial dysfunction [9], apoptosis and oxidative stress [10]. Increasing GV may contribute to eye (i.e., development and progression of retinopathy), renal (risk of nephropathy and albuminuria) and cardiovascular complications (i.e., cardiac autonomic neuropathy, acute coronary syndrome and stroke functional outcome) [11,12,13,14]. Thus, our previous study investigated the prognostic value of GV in patients with diabetes and acute coronary syndrome (ACS). Multivariable logistic regression analysis showed that GV > 2.70 mmol/L, a Synergy between PCI with Taxus and Cardiac Surgery (SYNTAX) score of >34 and reduced left ventricular ejection fraction of <40% were independent predictors of MACE, with odds ratios (ORs) of 2.21 (95% CI 1.64–2.98; *p* < 0.001), 1.88 (1.26–2.82; *p* = 0.002) and 1.71 (1.14–2.54; *p* = 0.009). One-fourth of these patients (24.4%) presented with a Killip score ≥ 2 [13]. In the context of AHF, three studies [15,16,17] already described an association between early GV and short-term MACE in patients with or without diabetes. Currently, a study dedicated to AHF diabetic patients evaluating the association between early GV and mid-term MACE occurrence is missing.

The aim of this study was to investigate the relationship between GV and mid-term MACE in diabetic patients with acute heart failure.

## 2. Materials and Methods

### 2.1. Study Population

In total, 1605 consecutive patients with decompensated HF were admitted to the Cardiac Emergency Department of the Bordeaux University Hospital between January 2015 and November 2016. Patients were selected using the following inclusion criteria: (i) confirmed diagnosis of AHF; (ii) intensive cardiac care unit (ICCU; level 3; tertiary referral unit) admission criteria, i.e., high-risk patients (with marked dyspnoea, haemodynamic instability, recurrent arrhythmias, AHF and associated acute coronary syndrome (ACS)) and/or any of the following criteria: need for mechanical ventilation (or already intubated), signs/symptoms of hypoperfusion, oxygen saturation (SpO_2_) < 90%, respiratory rate > 25/min, heart rate < 40 or >130 bpm, systolic blood pressure < 90 mmHg; (iii) glucose at admission < 16.7 mmol/L; (iv) confirmed diagnosis of type 1 or type 2 diabetes mellitus. Exclusion criteria were: (i) diabetic ketosis or nonketotic hyperosmolar coma at admission; (ii) stress hyperglycaemia during ICCU stay. Complete data were recorded in the hospital using DxCare^®^ software (Medasys), including previous history of cardiac disease, common comorbid conditions, clinical examination, biological data, extent of coronary artery disease (CAD) on invasive coronary angiography, AHF management and therapy strategies for diabetes. AHF and ACS were both defined and treated according to the European Society of Cardiology guidelines [18,19,20]. Type 1 or type 2 diabetes mellitus was diagnosed based on classical criteria including the previous use of insulin or glucose-lowering medication before hospitalisation and/or if HbA_1C_ ≥ 6.5% [21]. Stress hyperglycaemia was defined as a transient elevation of blood glucose of >198 mg/dL (11 mmol/L) due to the stress of disease. When a coronary angiography was performed during the index hospitalisation, significant CAD was reported as ≥50% stenosis in a major coronary vessel. QCA was used to validate the severity of each stenosis. CAD severity was determined as: no significant stenosis, one diseased vessel, two diseased vessels or left main and/or three-vessel disease (Section A.1). Echocardiography was performed at Bordeaux University Hospital during the ICCU stay to calculate the left ventricular ejection fraction (LVEF).

### 2.2. Capillary Blood Glucose Values

Blood glucose measurements were carried out using the ACCU-CHEK Inform II^®^ system (Roche Diagnostics GmbH, Mannheim, Germany). Blood glucose values were collected in a capillary mode throughout the stay, including the ICCU and conventional cardiology unit. The lab unit of the hospital assumed the management of glucometers, which were all connected to the central middleware cobas^®^ IT1000. Results were automatically transferred to the patient file. A quality program was established to offer the best quality of results. Linearity and repeatability tests were performed on devices before their distribution, and daily quality controls (QC) were carried out in the care services by the nurses. The point-of-care team of the lab supervised the analytical performances, the results of QC and the empowerment of healthcare operators whose access to devices was nominative.

### 2.3. Measurement of Glycaemic Variability

GV was computed from a discontinuous glucose profile. In view of implementing this method in everyday life, we arbitrarily chose to use standard deviation of glycaemia (mg/dL or mmol/L) instead of mean amplitude of glycaemic excursions (MAGE). GV was divided into three tertiles.

### 2.4. Diabetes Care

In ICCU, intravenous insulin therapy was delivered if necessary to target the glycaemic value between 7.7 and 10 mmol/L [22]. Continuous insulin administration was initiated when blood glucose on admission was ≥180 mg/dL (10.0 mmol/L) and/or when pre-meal glycaemia was ≥140 mg/dL (7.7 mmol/L) during the ICCU stay. During hospitalisation in non-intensive care wards, diabetes was managed following ESC guidelines with specialist advice if necessary [23].

### 2.5. Outcomes

The follow-up period was defined as the time elapsed between May and August 2020. The incidence of MACE was reported, including unplanned hospitalisation for heart failure, new-onset myocardial infarction, ischaemic stroke and cardiac death. Outcome data were collected using the following procedure: 1. consulting the medical records available in Bordeaux University Hospital; 2. contacting the patients’ general practitioners or cardiologists; 3. contacting the patients themselves. All MACE data were reviewed by at least one physician.

### 2.6. Statistical Analysis

All data collected were then anonymised and used in the analysis reported herein. Data are presented as frequencies or percentages for categorical variables, while continuous variables are presented as mean and standard deviation (SD) plus median and interquartile range (IQR) for abnormal distributed parameters. The distribution of the data was tested for normality to choose the adequate test (parametric or non-parametric). Categorical variables were compared using an χ² test with Yates correction. The relationships between GV and other variables were investigated using a linear regression analysis. A Pearson’s correlation coefficient of 0.40 to 0.69 indicates strong positive relationship and an r value of 0.30 to 0.39 indicates moderate positive relationship. Receiver operating characteristics (ROC) curve analyses were obtained in order to determine the optimal cut-off values for GV, admission glycaemia, mean glycaemia, B-type natriuretic peptide (BNP) value and the estimated glomerular filtration rate (eGFR) to predict MACE. Thus, the best cut-off values were used to binarise each variable for further multivariate analysis. Two groups were obtained according to the eGFR binarised value (< or ≥50 mL per minute per 1.73 m^2^ of body surface area) and the level of BNP (≤ or >615 pg/mL). HBA_1C_, LVEF and GV were also included as continuous and categorised variables (HbA_1C_ < 6.5% (48 mmol/mol) and ≥6.5% (48 mmol/mol); LVEF: <30% and ≥30%; LVEF: <40% and ≥40%; GV: ≤50 mg/dL (or ≤2.70 mmol/L) and >50 mg/dL (or >2.70 mmol/L)).

Univariate analysis was performed initially. Kaplan–Meier survival curves were used to represent the proportional risk of MACE for GV and the log-rank test was performed to assess differences between high levels and low levels of GV. To ascertain the independent contribution of GV, hypoglycaemia, admission glycaemia and mean glycaemia to MACE, and because these parameters were correlated, a Cox proportional-hazards regression analysis was carried out using several models including pre-defined and more relevant variables with a significance level of *p* < 0.15 in univariate analysis. To avoid bias due to too-small number of events per variable in proportional hazards analysis [24], a number of events per variable of <10 were chosen. ROC curve analysis determined that the best cut-off value was 47.5 mg/dL (2.61 mmol/L) for GV. The authors decided to test GV in the multivariate analysis as the following binarised value: GV: ≤50 mg/dL and >50 mg/dL, because this simple value, which is close to 47.5 mg/dl, was previously determined in a study dedicated to ACS [13]. In addition, there was no established cut-off value in AHF so far. Furthermore, data regarding the presence and recent updated extent of CAD were not available in many patients. This fact led the authors to exclude this parameter from the multivariate analysis. Hazard ratios (HRs) and 95% confidence intervals (CIs) were calculated. A *p* value of <0.05 was considered statistically significant. All statistical analyses were performed using NCSS software (NCSS 2001 Statistical software, Kaysville, UT, USA) and Kaplan–Meier event-free survival curves were constructed using SAS Software (SAS Institute, Cary, NC, USA, Version 9.4).

## 3. Results

### 3.1. Study Population

A total of 1605 patients with confirmed decompensated HF were referred to the Cardiac Emergency Department of Bordeaux University Hospital between January 2015 and November 2016. Among them, 400 diabetic patients were hospitalised in ICCU. Eight patients were lost during follow-up. Complete data of 392 patients were available for the final analysis (Figure 1). Table 1 shows the baseline characteristics of the study population. The median duration of ICCU stay was 3 days IQR (1–5). The median duration of hospitalisation was 10 days IQR (6–15). A total of 36 (9.2%) in-hospital deaths were reported. In total, 317 patients (80.9%) received intravenous insulin therapy in the ICCU to reach the target blood sugar threshold defined previously. A total of 16 patients (4.1%) were maintained between the two capillary blood glucose values of 140 mg/dL (or 7.7 mmol/L) and 180 mg/dL (or 10 mmol/L) in ICCU.

Among all glycaemia measurements collected in the study population (*n* = 20,141), glycaemia values were below 54 mg/dL (or 3 mmol/L) 211 times (1.0%), between 54 mg/dL (3 mmol/L) and 140 mg/dL (7.7 mmol/L) 7577 times (37.6%), between 140 mg/dL (7.7 mmol/L) and 180 mg/dL (10 mmol/L) 4978 times (24.7%) and ≥180 mg/dL (10 mmol/L) 7375 times (36.6%), respectively. Nine patients (2.3%) had newly diagnosed diabetes. Diabetes was well-controlled in the majority of patients (mean HbA1C = 7.38 ± 1.48%) as well as dyslipidaemia (mean LDL-C = 96 ± 48 mg/dL). Regarding GV (SD), mean value was 53 mg/dL (2.90 mmol/L) (SD); GV tertiles were <43 mg/dL (2.37 mmol/L), between 43 mg/dL (2.37 mmol/L) and 58 mg/dL (3.23 mmol/L) and >58 mg/dL (3.23 mmol/L), respectively. GV (SD) was significantly higher in the insulin therapy group as compared to the group without insulin therapy in ICCU (57 versus 44 mg/dL (or 3.14 versus 2.42 mmol/L); *p* < 0.001). The associations of GV with admission glycaemia and HbA1c were significant (Pearson’s correlation coefficient r = 0.453 and r = 0.387, respectively; all *p* < 0.001). The association of GV with hypoglycaemia encountered at any time during the hospitalisation per individual patient was significant (Pearson’s correlation coefficient r = 0.481; *p* < 0.001). Conversely, the correlation of GV with eGFR was not significant (Pearson’s correlation coefficient r = −0.091; *p* = 0.07).

### 3.2. Incidence of MACE

Global median follow-up time was 29 months IQR (6–51). MACE occurred in 227 patients (57.9%): 92 patients (23.5%) died of cardiac causes, 107 patients (27.3%) were hospitalised for heart failure, 19 (4.8%) had new-onset myocardial infarction and 9 (2.3%) had an ischaemic stroke. At 1 year, death from cardiovascular causes had occurred in 72 of the 392 patients (18.4%). The combined endpoint of mortality or unplanned HF hospitalisation within 1 year had a rate of 34.4% (135 out of 392).

### 3.3. Causes of Cardiovascular and Non-Cardiovascular Death

Overall, 131 deaths (33.4%) occurred during follow-up (Table 2). Cardiovascular death was identified in 92 patients, whereas 39 non-CV deaths (9.9%) occurred. Heart failure was the most common CV cause of death (50 out of 131 patients (38.2%)). The remainder died mainly from arrhythmias and ischaemic events. Regarding non-CV deaths, cancer was the most common cause, identified in 15 patients.

### 3.4. Univariate Regression Analysis

In univariate analysis, the following parameters were associated with MACE occurrence: female gender, age, age > 75 years, eGFR < 50 mL/min/1.73 m^2^, LVEF < 30%, B-type natriuretic peptide (BNP) > 615 pg/mL, left main and/or three-vessel disease, SYNTAX score > 33, GRACE score > 140, acute kidney failure during hospitalisation, cardiogenic shock during hospitalisation, use of vasopressor/inotropic agents, mechanical ventilation during hospitalisation, renal replacement therapy, admission glucose level, mean glycaemia, percentages of hypoglycaemia and hyperglycaemia, GV (SD) values (except for first and second GV (SD) tertiles), level of glycaemia per patient and length of hospital stay (number of days) (Table 3).

### 3.5. Multivariate Regression Analysis

The variables included were age > 75 years, female sex, LVEF < 30%, BNP value > 615 pg/mL, eGFR < 50 mL/min/1.73 m^2^, acute kidney failure during hospitalisation, mechanical ventilation during hospitalisation, use of vasopressor/inotropic agents, GV (SD) > 50 mg/dL (or 2.70 mmol/L) and length of hospital stay. Multivariate Cox proportional-hazards regression analysis (Table 4) showed that GV > 50 mg/dL (or 2.70 mmol/L), age > 75 years, reduced left ventricular ejection fraction (LVEF < 30%) and female sex increased the risk of MACE by 3.16 (2.25–4.43; *p* < 0.001), 1.54 (1.14–2.08; *p* = 0.005), 1.47 (1.06–2.07; *p* = 0.02) and 1.43 (1.05–1.94; *p* = 0.03), respectively. AHF diabetic patients with a higher GV level (>50 mg/dL or 2.70 mmol/L) had a significantly higher incidence of MACE: cardiac mortality (*p* < 0.001), hospitalisation for heart failure (*p* < 0.001), new-onset myocardial infarction (*p* < 0.03), stroke (*p* < 0.04) and combined MACE (*p* < 0.001). AHF patients older than 75 years had a significantly higher incidence of MACE: cardiac mortality (*p* < 0.001), hospitalisation for heart failure (*p* < 0.001) and combined MACE (*p* < 0.001). Concerning hospitalisation for new-onset myocardial infarction and stroke, there was no significant difference in adverse cardiovascular event rates between the two study groups (*p* = 0.74 and *p* = 0.07, respectively). AHF patients with an LVEF < 30% had a significantly higher incidence of cardiac mortality (*p* = 0.002), of hospitalisation for heart failure (*p* < 0.001), of stroke (*p* = 0.02) and of all combined MACE (*p* < 0.001). Concerning new-onset myocardial infarction, there was no significant difference in adverse cardiovascular event rates between the two study groups (*p* = 0.24). AHF female patients with diabetes had a significantly higher incidence of MACE: cardiac mortality (*p* = 0.004), hospitalisation for heart failure (*p* = 0.03), new-onset myocardial infarction (*p* = 0.02) and combined MACE (*p* = 0.001). Concerning hospitalisation for acute stroke, there was no significant difference in adverse cardiovascular event rates between the two study groups (*p* = 0.14). Kaplan–Meier event-free survival curves for freedom from MACE in the two patient groups according to admission GV level are shown in Figure 2. Kaplan–Meier survival curves for cardiovascular death in the two patient groups according to admission GV level are shown in Figure 3. Association of hypoglycaemia, admission glycaemia and mean glycaemia with MACE are detailed in Section A.2.

## 4. Discussion

This study explored the association between GV, well-known cardiovascular risk factors, established cardiac parameters and mid-term MACE in diabetic patients with AHF. Our results show that elevated GV (SD, >50 mg/dL or 2.70 mmol/L) was the strongest independent predictive factor for increased risk of mid-term MACE in this population. Furthermore, age > 75 years, reduced LVEF < 30% and female sex were also independent predictive factors for MACE.

Our study, which focused on a particular population of diabetic patients during 29 months IQR (6–51), expands on previous studies showing the importance of GV in predicting prognosis of patients admitted for AHF. Dungan et al. [15] previously reported an association between high GV and in-hospital death in patients hospitalised for AHF. However, only 36% of their population had diabetes and their status remained undetermined since the majority of AHF patients were admitted in a heart hospital, where beds were exchangeable between ICCU and non-ICCU status. In the same way, Lazzeri et al. [16] documented that early GV (measured in the first 24 h from ICCU admission) was an independent predictor of mortality at a mean follow-up of 10.4 months. However, this study, which included 247 consecutive patients, did not collect the diabetic status. The results of our study seem to be credible as the patients were managed according to the guidelines for common cardiovascular risk factors (mean LDL-C = 2.48 mmol/L, mean HbA1C = 7.38%) before hospitalisation. This “real-life” and all-comer diabetic AHF patient population, which included patients with and without history of chronic HF, with mild to moderately impaired renal function at baseline, with reduced LVEF and preserved LVEF, with and without insulin use, is a very representative cohort. Medical treatment at hospital discharge was suboptimal and comparable to the Observatoire Français de l’Insuffisance Cardiaque Aigue (OFICA) survey [26]. In-hospital and 1-year mortalities are similar to previous registries [26,27]. Our study did not show the current trends in modes of death in heart failure: patients with HF die less due to sudden death and more due to non-CV causes, mainly cancer [28]. Baseline severe patient characteristics may explain these results: 52 out of 392 patients (13.3%) presented with cardiogenic shock in our tertiary referral unit, whereas 159 out of 392 patients (40.6%) were admitted for an acute coronary syndrome. Furthermore, our pilot study reflects a “real-life” population as it included all consecutive diabetic patients hospitalised with AHF during the study period. Interestingly, the RR for GV > 50 mg/dL was superior to that for age > 75 years, reduced LVEF and female sex, three well-known cardiovascular parameters associated with the occurrence of MACE [29,30,31]. In addition, GV seems to be a better predictive factor of mid-term MACE than worsening renal failure and BNP in diabetic patients with AHF, whereas these markers usually provide robust prognostic information in all patients with AHF [32,33]. There is still extensive debate about GV as a major risk factor for the development of CAD and other vascular complications, in parallel with the search for possible cellular mechanisms. Our team has previously shown that a GV cutoff value of >2.70 mmol/L was the strongest independent predictive factor for midterm MACE in patients with diabetes and ACS [13]. Another study observed an interesting correlation between GV and left ventricular remodelling [34]. High GV might reflect higher activation of the neurohormonal system, which is known to be deleterious in HF. The proposed mechanisms underlying the deleterious effects of GV also include short-term fluctuations of glycaemia leading to endothelial dysfunction, apoptosis and oxidative stress [9]. Many methods have been proposed for measuring short- and long-term GV, but there is no universally accepted “gold standard”. For short-term (24 h) GV, MAGE and SD are among the most widely used and seem to be efficient [4,8]. The preferred measure of GV is the coefficient of variation for glucose (%CV) which is defined as the SD adjusted to the 24 h mean glucose concentration. A cut-off value of 36% was previously validated [35]. In our study, using discontinuous glucose monitoring and defining the %CV cut-off value as the SD cut-off value (50 mg/dL or 2.70 mmol/L) adjusted to the mean glucose concentration (145 mg/dL or 8.0 mmol/L) during hospitalisation, we computed a %CV cut-off value of 34%. This result is in accordance with a recent published study showing that a %CV > 30.0% during AHF hospitalisation increased the risk of mortality by 2.21 (1.16–4.21; *p* = 0.02) in the following 6 months [17]. Unfortunately, regular postprandial glycaemia measurements were lacking in this study.

Today, current guidelines concerning diabetes management of AHF patients [22] propose that insulin-based glycaemic control should be considered in cases of hyperglycaemia (>10 mmol/L or >180 mg/dL) with the target adapted to possible comorbidities. Exact targets are still to be determined. Lanspa et al. studied 6101 critically ill adults supported with eProtocol-insulin. %CV was independently associated with 30-day mortality (odds ratio 1.23 for every 10% increase, *p* < 0.001), even after adjustment for hypoglycaemia, age, disease severity and comorbidities [36]. Thus, reducing short-term GV may become an objective in the acute phase. Inhibitors of sodium–glucose cotransporter 2 (iSGLT2) could provide a new therapeutic strategy by reducing GV [37,38], which could explain their cardiac benefits on HF [39]. Finally, to the best of our knowledge, no study has established the impact of decreasing short-term GV in the context of AHF.

### Study Limitations

Higher variabilities of metabolic parameters might be observed in patients with generalised frailty [40]. Patients with more pronounced and frequent swings in glucose levels are probably the ones with increased co-morbidities and/or who are more exposed to cardiovascular complications. To provide a risk score for the general risk, including cardiovascular risk, would be very useful. Nevertheless, such a universally accepted general risk score does not currently exist. The choice of SD (which reflects more dispersion than variability) is debatable [4]. However, some studies suggested that the random sampling errors in SD are significantly and consistently smaller than in other variables such as MAGE [41]. SD is simple and probably efficient to evaluate GV and its evolution in daily clinical practice [41,42]. Continuous glucose monitoring (CGM) was not inserted in the patients of our study. Thus, to equip all consecutive AHF patients (more than 1600 patients in the cardiac emergency department) with an implantable device under the skin is difficult in the emergency setting (most of them need a calibration and therefore stable glucose). Furthermore, possible variations in subcutaneous glucose recovery due to haemodynamic alterations (i.e., hypotension, shock, vasoactive drugs, bleeding consecutive to antithrombotic regimen) could alter CGM signal. In addition, a real-time CGM device is not approved in Europe and USA to adjust the insulin dose in ICCU. Lastly, it would have been interesting to depict the fluctuations of GV and common cardiovascular risk factors during follow-up.

## 5. Conclusions

Glycaemic variability (i.e., SD > 50 mg/dL or 2.70 mmol/L) is a powerful independent predictive factor of mid-term major adverse cardiac events in diabetic patients hospitalised with acute heart failure. A high GV must at least alert physicians in charge of patients on their potential cardiovascular risk. Whether reducing short-term GV can decrease the incidence of MACE is still an unresolved issue. Nevertheless, this study emphasises that a high GV should probably be avoided in diabetic patients with AHF.

## Figures and Tables

**Figure 1 jcm-11-01549-f001:**
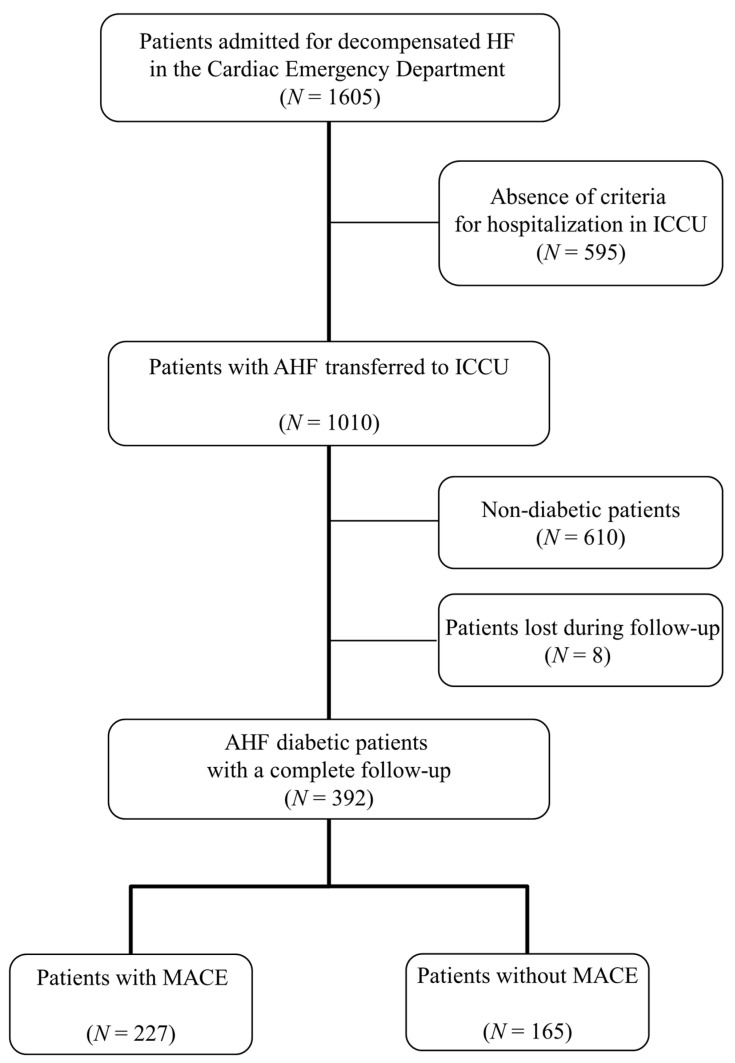
Flowchart of the study. AHF: acute heart failure; ICCU: intensive cardiac care unit; HF: heart failure; MACE: major cardiovascular events.

**Figure 2 jcm-11-01549-f002:**
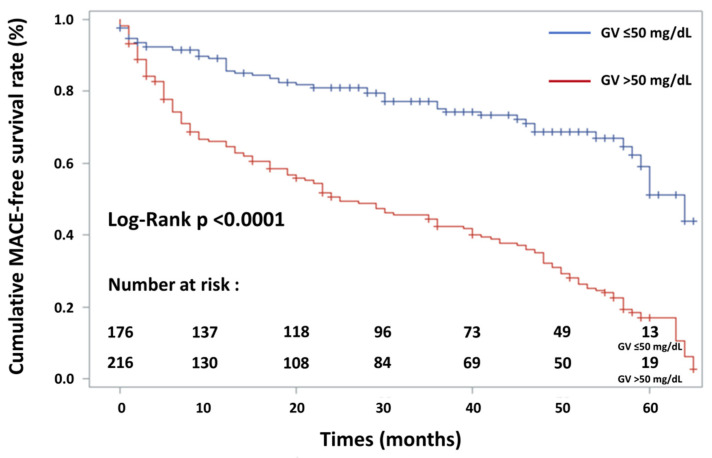
Kaplan–Meier event-free survival curves for freedom from MACE in two patient groups according to glycaemic variability (GV). Event-free survival rate was significantly lower in patients with high GV (log-rank test, *p* < 0.0001—solid blue line: GV ≤ 50 mg/dL; solid red line: GV > 50 mg/dL).

**Figure 3 jcm-11-01549-f003:**
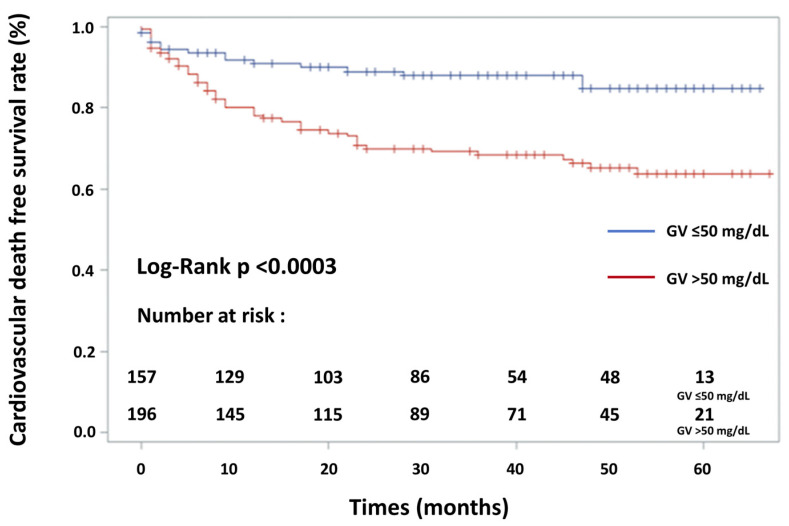
Kaplan–Meier cardiovascular death-free survival curves for the patients in two groups according to glycaemic variability (GV). Pairwise comparisons between patient groups are given. Survival rate was significantly lower in patients with high GV (log-rank test, *p* < 0.0003—solid blue line: GV ≤ 50 mg/dL; solid red line: GV > 50 mg/dL).

**Table 1 jcm-11-01549-t001:** Patients’ characteristics and management.

Baseline Characteristics	*N* = 392
Age (years)	73 ± 10.2
Age > 75 (years)	191 (48.7)
Female sex (%)	121 (30.9)
Prior HF hospitalisation	125 (31.9)
History of chronic heart failure	214 (54.6)
Systolic blood pressure at admission (mmHg)	131 ± 41.2
Heart rate at admission (beats/min)	87 ± 25.9
Smoking status (%)	
Non-smoker	224 (57.1)
Former smoker	126 (32.2)
Current smoker	42 (10.7)
Hypertension *	329 (83.9)
Type of diabetes	
Type 1	12 (3.1)
Type 2	378 (96.4)
Secondary (chronic pancreatitis)	2 (0.5)
HbA_1c_ (%)	7.38 ± 1.48
Total cholesterol (mg/dL)	159 ± 57
LDL cholesterol (mg/dL)	96 ± 48
HDL cholesterol (mg/dL)	42 ± 16
TG (mg/dL)	262 ± 209
BMI (kg/m^2^)	29.1 ± 6.5
Family history of CAD (%)	33 (8.4)
Personal history of CAD (%)	163 (41.6)
Atrial fibrillation	76 (19.4)
Chronic respiratory failure (%)	96 (24.5)
eGFR (mL/min/1.73 m^2^) †	55.1 ± 29.2
CKD with RRT (%)	8 (2.0)
Left ventricular ejection fraction (LVEF) (%)	43.3 ± 13.0
LVEF < 30%	81 (20.7)
LVEF 30–40%	72 (18.4)
LVEF 40–49%	95 (24.2)
LVEF ≥ 50%	144 (36.7)
Pre-existent aetiologies of cardiopathies predisposing to AHF	
Ischaemic heart disease	142 (36.2)
Toxic damage	19 (4.8)
Immune-mediated and inflammatory damage	2 (0.5)
Infiltration	2 (0.5)
Metabolic derangements	5 (1.3)
Genetic abnormalities	20 (5.1)
Valve and myocardium structural defects	102 (26.0)
Pericardial and endomyocardial pathologies	5 (1.3)
Tachycardia-induced cardiomyopathy	7 (1.8)
Prior HF hospitalisation in patient with preserved LVEF	36 (9.2)
No pre-existent cardiopathy	52 (13.3)
Factors triggering AHF	
Acute coronary syndrome (ACS)	159 (40.6)
Myocardial rupture complicating ACS ‡	3 (0.7)
Hypertensive emergency	40 (10.2)
Tachyarrhythmia	50 (12.8)
Bradyarrhythmia	12 (3.1)
Acute native or prosthetic valve incompetence §	27 (6.9)
Vigorous fluid administration	10 (7.7)
Non-adherence with salt/fluid intake or medications	32 (8.2)
Worsening renal failure	20 (5.1)
Severe anaemia ¶	20 (5.1)
Infection (e.g., pneumonia, sepsis)	76 (19.4)
Stress-related cardiomyopathy	3 (0.7)
Metabolic/hormonal derangements #	7 (1.8)
Toxic substances, cardiodepressant and other drugs **	5 (1.3)
BNP value at admission (pg/mL)	597 (348–1300)
Arterial pH at admission (*n* = 200)	7.35 ± 0.11
Arterial blood lactate at admission (mmol/L) (*n* = 200)	2.35 ± 2.57
Acute kidney failure †† (%)	145 (37.0)
Cardiogenic shock ‡‡	52 (13.3)
Peak troponin I (ng/mL) (normal < 0.04) §§	0.34 (0.09–6)
Presence and extent of CAD (%)	
No invasive angiography	174 (44.4)
No CAD	22 (5.6)
No significant stenosis	44 (11.2)
One-vessel disease	40 (10.2)
Two-vessel disease	44 (11.2)
Left main and/or three-vessel disease	68 (17.4)
SYNTAX score	20.0 ± 13.2
GRACE Score (*n* = 159)	168 ± 37
GRACE Score > 140	118 (74.2)
Management	
Oxygen therapy	385 (98.2)
Diuretics	375 (95.7)
Intravenous vasodilators	104 (26.5)
Inotropic agents ¶¶	65 (16.6)
Vasopressors ##	43 (11.0)
Non-invasive positive pressure ventilation	114 (29.1)
Mechanical ventilation	47 (12.0)
Duration of mechanical ventilation (days)	3.3 ± 5.6
Renal replacement therapy	34 (8.7)
Intra-aortic balloon pump	8 (2.0)
Mechanical circulatory support ***	10 (2.6)
Heart transplantation	2 (0.5)
Myocardial revascularisation	
PCI	126 (32.1)
CABG	23 (5.9)
Hybrid strategy	2 (0.5)
Medical treatment only †††	8 (2.0)
Electrical cardioversion	20 (5.1)
Ablation for arrhythmias	5 (1.2)
Pacemaker	14 (3.6)
Non-surgical device treatment of heart failure	
Cardiac resynchronisation therapy	9 (2.3)
Implantable cardioverter-defibrillator	5 (1.2)
Valvular heart disease treatment	
Trans-aortic valve replacement	14 (3.6)
Valve surgery	22 (5.6)
Treatment at hospital discharge	
Diuretics	301 (76.8)
RAASI	241 (61.5)
ARNi	5 (1.2)
ß-Blocker	260 (66.3)
Mineralocorticoid antagonists	124 (31.6)
Antithrombotic treatment (%)	
Single APT (SAPT)	77 (19.6)
DAPT	96 (24.5)
OAC monotherapy	75 (19.1)
Dual therapy (OAC + SAPT)	62 (15.8)
Triple therapy (OAC + DAPT)	27 (6.9)
Statin	269 (68.6)
Oral glucose-lowering therapies	197 (50.3)
Biguanides	116 (29.6)
Sulfonylureas and meglitinides	123 (31.4)
α-glucosidase inhibitors	5 (1.2)
DDP-4 inhibitors	49 (12.5)
GLP-1 receptor agonists	5 (1.2)
Insulin therapy	193 (49.2)
Glycaemic status	
Glycaemia assays per patient	45 (23–78)
Glycaemia assays per patient per day	5 (3–8)
Admission glycaemia (mg/dL)	193 (87)
Mean glycaemia (mg/dL)	164 (32)
Percentage of hypoglycaemia ‡‡‡ (%)	0.9
Number of patients with hypoglycaemia (%)	84 (21.4)
Hypoglycaemia events per patient, *n*	2 (1–2)
Percentage of hyperglycaemia §§§ (%)	32.6
Number of patients with hyperglycaemia (%)	357 (91.1)
Glycaemic variability (SD, mg/dL)	53 (22)

Data shown are number (%), median (25th–75th percentiles) or mean ± SD. ACS: acute coronary syndrome; AHF: acute heart failure; APT: antiplatelet therapy; ARNi: angiotensin receptor-neprilysin inhibitor; BNP: B-type natriuretic peptide; BMI: body mass index; CABG: coronary artery bypass graft surgery; CAD: coronary artery disease; CKD with RRT: chronic kidney disease to renal replacement therapy; DAPT: dual antiplatelet therapy; eGFR: estimated glomerular filtration rate; GRACE: Global Registry of Acute Coronary Events; HDL: high-density lipoprotein; HF: heart failure; LDL: low-density lipoprotein; LVEF: left ventricular ejection fraction; OAC: oral anticoagulation therapy; PCI: percutaneous coronary intervention; RAASI: renin-angiotensin-aldosterone system inhibitors; SAPT: single antiplatelet therapy; SYNTAX: Synergy between PCI with Taxus and Cardiac Surgery. TG: triglyceride. * Hypertension was defined as systolic blood pressure ≥ 140 mmHg and/or diastolic blood pressure ≥ 90 mmHg or treatment with oral antihypertensive drugs. † eGFR was calculated with the use of the simplified Modification of Diet in Renal Disease formula. ‡ Myocardial rupture complicating ACS (free wall rupture, ventricular septal defect, acute mitral regurgitation). § Acute native or prosthetic valve incompetence secondary to endocarditis, aortic dissection or thrombosis. ¶ Severe anaemia defined as haemoglobin level between 4 g/dL and 8 g/dL. # Metabolic/hormonal derangements defined as thyroid dysfunction, adrenal dysfunction or pregnancy and peripartum-related abnormalities. ** Toxic substances (alcohol, recreational drugs), cardiodepressant and other treatments (non-steroidal anti-inflammatory drugs, corticosteroids, negative inotropic substances, cardiotoxic chemotherapeutics). †† Acute kidney failure defined according to AKIN network (stage ≥ 1: absolute increase of serum creatinine 1.5–2.0 times from baseline or ≥0.3 mg/dL (≥26.5 µmol/L)). ‡‡ Cardiogenic shock was defined as hypotension (systolic blood pressure < 90 mmHg) despite adequate filling status associated with clinical and biological markers of hypoperfusion. §§ Troponin I assay was performed in biochemistry central lab on multiparametric automate Access/DXi 800 Beckman, as BNP. ¶¶ Inotropic agents: dobutamine, dopamine, levosimendan. ## Vasopressors: norepinephrine or epinephrine. *** Mechanical circulatory support includes Impella^®^ device and veno-arterial extracorporeal membrane oxygenation. ††† Medical treatment strategy or failure of revascularisation. ‡‡‡ Detection of glucose concentration < 54 mg/dL (or <3 mmol/L) among all measurements obtained in all patients at any time during hospitalisation. §§§ Detection of glucose concentration ≥ 180 mg/dL (or ≥10 mmol/L) among all measurements obtained in all patients at any time during hospitalisation.

**Table 2 jcm-11-01549-t002:** Rates of all-cause, cause-specific cardiovascular death and non-cardiovascular death.

All-Cause Death, *n* (%)	131 (33.4)
Cause-specific CV and non-CV death	
Cardiovascular cause of death, *n* (%)	92 (23.5)
Sudden cardiac death, *n* (%)	15 (3.8)
Heart failure/Cardiogenic shock, *n* (%)	50 (12.8)
Acute myocardial infarction, *n* (%)	16 (4.1)
Stroke, *n* (%)	1 (0.3)
Cardiovascular haemorrhage, *n* (%)	1 (0.3)
Cardiovascular procedure, *n* (%)	6 (1.5)
Other cardiovascular causes, *n* (%)	3 (0.8)
Non-cardiovascular cause of death, *n* (%)	39 (9.9)
Malignancy, *n* (%)	15 (3.8)
Infection (including sepsis), *n* (%)	13 (3.3)
Other, *n* (%)	10 (2.6)
Undetermined death, *n* (%)	1 (0.3)

All the causes of death were reported according to the ACC/AHA Key Data Elements and Definitions for Cardiovascular Endpoint Events in Clinical Trials [25].

**Table 3 jcm-11-01549-t003:** Univariate Cox proportional-hazards regression analysis for MACE.

Variables	Risk Ratio	95% CI	*p* Value
Age, years	1.02	1.01–1.04	**<0.001**
Age > 75 years	1.27	1.11–1.45	**<0.001**
Female sex	1.17	1.01–1.35	**0.03**
Prior HF hospitalisation	1.04	1.01–1.08	0.07
History of chronic heart failure	0.95	0.87–1.05	0.31
Current smoker status	0.92	0.71–1.20	0.10
Hypertension	0.88	0.74–1.05	0.15
Diabetes type	0.61	0.25–1.47	0.27
HbA_1c_ ≥ 6.5%	1.03	0.64–1.50	0.19
TC (mg/dL)	0.80	0.56–1.15	0.23
LDL-C (mg/dL)	0.89	0.57–1.38	0.60
HDL-C (mg/dL)	0.78	0.23–2.67	0.70
TG (mg/dL)	0.92	0.71–1.19	0.53
BMI (kg/m^2^)	0.99	0.97–1.02	0.46
eGFR < 50 mL/min/1.73 m^2^	1.78	1.36–2.34	**<0.001**
CKD with RRT	2.22	0.98–5.03	0.06
Family history of CAD	1.00	0.85–1.14	0.96
Personal history of CAD	1.09	0.95–1.25	0.21
Atrial fibrillation	1.07	0.86–1.34	0.53
Chronic respiratory failure	1.05	0.89–1.22	0.58
Systolic blood pressure at admission (mmHg)	1.00	0.99–1.00	0.14
Heart rate at admission (beats/min)	1.00	0.99–1.00	0.57
LVEF (%)	0.99	0.98–1.00	0.10
LVEF of <30% (compared with LVEF of ≥30%)	1.51	1.11–2.05	**0.009**
LVEF of <40% (compared with LVEF of ≥40%)	1.08	0.94–1.24	0.28
Factors triggering acute heart failure			
Acute coronary syndrome (ACS)	0.91	0.80–1.05	0.20
Myocardial rupture complicating ACS	0.66	0.40–1.10	0.11
Hypertensive emergency	1.00	0.94–1.07	0.94
Tachyarrhythmia	1.01	0.93–1.08	0.88
Bradyarrhythmia	1.01	0.96–1.05	0.52
Acute native or prosthetic valve incompetence	1.05	1.01–1.10	0.06
Vigorous fluid administration	1.00	0.98–1.02	0.75
Non-adherence with salt/fluid intake or medications	1.02	0.97–1.07	0.45
Worsening renal failure	1.02	0.98–1.05	0.63
Severe anaemia	1.01	0.94–1.07	0.50
Infection (e.g., pneumonia, sepsis)	1.03	0.88–1.21	0.58
Stress-related cardiomyopathy	1.01	0.94–1.09	0.76
Metabolic/hormonal derangements	1.01	0.98–1.03	0.46
Toxic substances, cardiodepressant and other drugs	1.03	1.00–1.05	0.07
BNP value at admission > 615 pg/mL	1.84	1.40–2.42	**<0.001**
Arterial pH at admission	0.96	0.20–4.72	0.96
Arterial blood lactate at admission (mmol/L)	0.99	0.91–1.08	0.89
Acute kidney failure during hospitalisation	1.65	1.25–2.17	**<0.001**
Cardiogenic shock	1.87	1.30–2.70	**<0.001**
Peak troponin (ng/mL)	1.00	0.99–1.01	0.38
Extent of CAD			
One-vessel disease	0.67	0.45–1.10	0.94
Two-vessel disease	0.85	0.39–1.47	0.58
Left main and/or three-vessel disease	1.37	1.10–1.70	**0.004**
SYNTAX Score ≤ 22	1.12	0.84–1.49	0.45
22 < SYNTAX Score ≤ 33	1.15	0.96–1.35	0.13
SYNTAX Score > 33	1.73	0.99–3.01	**0.04**
GRACE score	0.99	0.99–1.00	0.71
GRACE score > 140	1.98	1.16–3.40	**0.01**
Length of stay (number of days)	1.00	1.00–1.01	**0.04**
Management			
Oxygen therapy	0.92	0.64–1.33	0.66
Diuretics	0.78	0.45–1.34	0.36
Intravenous vasodilators	0.70	0.44–1.10	0.11
Vasopressors/inotropic agents	1.85	1.31–2.62	**<0.001**
Non-invasive positive pressure ventilation	0.93	0.70–1.24	0.61
Mechanical ventilation	1.56	1.03–2.38	**0.04**
Duration of mechanical ventilation (days)	1.05	1.02–1.09	**0.006**
Renal replacement therapy	1.84	1.12–3.03	**0.02**
Intra-aortic balloon pump	0.83	0.27–2.62	0.76
Mechanical circulatory support	4.30	0.60–30.7	0.15
Myocardial revascularisation			
PCI	1.07	0.86–1.34	0.53
CABG	1.11	0.40–3.07	0.85
Hybrid strategy	1.19	0.59–2.39	0.97
Medical treatment only	1.03	0.49–2.22	0.63
Glycaemia assays per patient	1.00	1.00–1.01	**0.02**
Admission glucose level (mg/dL)	1.00	1.00–1.01	**0.008**
Mean glycaemia (mg/dL)	1.01	1.00–1.01	**0.03**
Hypoglycaemia (%)	1.84	1.36–2.47	**<0.001**
Hyperglycaemia (%)	3.10	1.72–5.58	**<0.001**
GV (SD, mg/dL)	1.02	1.01–1.04	**<0.001**
GV > 2.70 (SD, mg/dL)	5.01	1.78–3.01	**<0.001**
GV tertiles			
First	1.23	0.39–3.88	0.72
Second	1.70	0.45–6.35	0.43
Third	6.22	2.08–18.6	**<0.001**

CI: confidence interval; SD: standard deviation; *p* values in boldface type indicate numbers that are significant at the 95% confidence limit.

**Table 4 jcm-11-01549-t004:** Multivariate Cox proportional-hazards regression analysis for MACE.

Variables	Risk Ratio	95% CI	*p* Value
Age > 75 years	1.54	1.14–2.08	**0.005**
Female sex	1.43	1.05–1.94	**0.03**
LVEF of <30% (compared with LVEF of ≥30%)	1.47	1.06–2.07	**0.02**
BNP value at admission > 615 pg/mL	1.30	0.95–1.77	0.09
eGFR < 50 mL/min/1.73 m^2^	1.10	0.80–1.52	0.54
Acute kidney failure during hospitalisation	1.27	0.92–1.76	0.15
Mechanical ventilation	1.23	0.70–2.16	0.48
Vasopressor/inotropic agents	1.60	0.97–2.64	0.06
GV > 50 mg/dL (or >2.70 mmol/L)	3.16	2.25–4.43	**<0.001**
Length of stay (days)	1.00	0.99–1.01	0.83

CI: confidence interval. *p* values in bold indicate numbers that are significant at the 95% confidence limit.

## Data Availability

The datasets supporting the conclusions of this article are available from the corresponding author on reasonable request.

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
