# Peer review of "Glycaemic Variability and Hyperglycaemia as Prognostic Markers of Major Cardiovascular Events in Diabetic Patients Hospitalised in Cardiology Intensive Care Unit for Acute Heart Failure"

_jcm, 2022, doi:10.3390/jcm11061549_

Round 1

Reviewer 1 Report

This article is very interesting. I think it is appropiated for publication in JCM.

Reviewer 2 Report

The manuscript by Gerbaud et al. is an observational study on the prognostic value of glycemic variability in acute heart failure mid-term outcomes. I commend the Authors for their nice work and my comments are mostly minor.

  • Line 52,53. I would specify what type of association you referred to (e.g., positive association with poor outcomes, inverse relationship etc.)
  • Line 59-60 Please specify that you referred to diabetes-associated complications
  • Line 62-64 Please rephrase the explanation on the studies you mentioned since it is unclear and not very readable. I would also move this section in the discussion
  • Please be consistent throughout the text with the units of measure (e.g., glucose sometimes is expressed by mmol/L only, sometimes with mg/dl and (mmol/L).
  • Since you do not only analyze glycemic variability, I would probably use a more general title (e.g., Hyperglycemia and glycemic variability as prognostic markers of MACE in diabetic patients hospitalized in Cardiology Intensive Care Unit for Acute heart failure”

Reviewer 3 Report

The authors examined 392 consecutive cases with diabetes and acute heart failure (AHF), and showed glycemic variability (GV) was an independent predictive factor of mid-term major cardiovascular events (MICE) in the patients. The study is closely related the author’s previous study (Diabetes Care. 2019;42:674-681), but the authors did not properly described it in introduction and discussion.

The reviewer has several comments.

Major comments

  1. Author’s previous study (Diabetes Care. 2019;42:674-681) examined consecutive patients with diabetes and acute coronary syndrome between January 2015 and November 2016 (same period), and showed GV was an independent predictive factor of mid-term MICE in the patients. Patients with and left ventricular ejection fraction < 40% had a significantly higher incidence of cardiac death, hospitalization for AHF, and MICE. However, the authors described “Nevertheless, the potential effect of glycaemic variability (GV) on mid-term major cardiovascular events (MACE) in diabetic patients remains uncertain.” in abstract. There is a lack of detailed description of the previous in introduction. There is a lack of discussion on GV and AHF in diabetes. The objectivity is not guaranteed. I have a strong concern on the author’s stance on the research.
  2. Incidence of MICE. The study showed high mortality rate in diabetes with AHF. One-fourth of the patients died of cardiac causes. Please show a figure of survival rate of the all patients and two groups (GV > 50 mg/dL, ≤ 50 mg/dL) and a table of causes of death, and discuss them.
  1. Are there any difference between patients with GV> 50 mg/dL and GV ≤ 50 mg/dL on clinical and laboratory backgrounds and severity of the underlying cause of AHF.
  2. Minor comment. Table 1. Glycemic status. Number of glycemia per patient?, Number of glycaemia per patient per day?

Round 2

Reviewer 3 Report

I have no comment for the revised version.